# Machine Learning Based on Event-Related EEG of Sustained Attention Differentiates Adults with Chronic High-Altitude Exposure from Healthy Controls

**DOI:** 10.3390/brainsci12121677

**Published:** 2022-12-07

**Authors:** Haining Liu, Ruijuan Shi, Runchao Liao, Yanli Liu, Jiajun Che, Ziyu Bai, Nan Cheng, Hailin Ma

**Affiliations:** 1Psychology Department, Chengde Medical University, Chengde 067000, China; 2Hebei Key Laboratory of Nerve Injury and Repair, Chengde Medical University, Chengde 067000, China; 3Hebei International Research Center of Medical Engineering, Chengde Medical University, Chengde 067000, China; 4Plateau Brain Science Research Center, Tibet University/South China Normal University, Lhasa 850012, China; 5Department of Biomedical Engineering, Chengde Medical University, Chengde 067000, China

**Keywords:** high-altitude chronic hypoxia, Go/No-Go, sustained attention, EEG, inhibitory control, machine learning

## Abstract

(1) Objective: The aim of this study was to examine the effect of high altitude on inhibitory control processes that underlie sustained attention in the neural correlates of EEG data, and explore whether the EEG data reflecting inhibitory control contain valuable information to classify high-altitude chronic hypoxia and plain controls. (2) Methods: 35 chronic high-altitude hypoxic adults and 32 matched controls were recruited. They were required to perform the go/no-go sustained attention task (GSAT) using event-related potentials. Three machine learning algorithms, namely a support vector machine (SVM), logistic regression (LR), and a decision tree (DT), were trained based on the related ERP components and neural oscillations to build a dichotomous classification model. (3) Results: Behaviorally, we found that the high altitude (HA) group had lower omission error rates during all observation periods than the low altitude (LA) group. Meanwhile, the ERP results showed that the HA participants had significantly shorter latency than the LAs for sustained potential (SP), indicating vigilance to response-related conflict. Meanwhile, event-related spectral perturbation (ERSP) analysis suggested that lowlander immigrants exposed to high altitudes may have compensatory activated prefrontal cortexes (PFC), as reflected by slow alpha, beta, and theta frequency-band neural oscillations. Finally, the machine learning results showed that the SVM achieved the optimal classification F1 score in the later stage of sustained attention, with an F1 score of 0.93, accuracy of 92.54%, sensitivity of 91.43%, specificity of 93.75%, and area under ROC curve (AUC) of 0.97. The results proved that SVM classification algorithms could be applied to identify chronic high-altitude hypoxia. (4) Conclusions: Compared with other methods, the SVM leads to a good overall performance that increases with the time spent on task, illustrating that the ERPs and neural oscillations may provide neuroelectrophysiological markers for identifying chronic plateau hypoxia.

## 1. Introduction

The plateau environment frequently causes nervous system damage due to many factors, such as low oxygen, low pressure, low temperature, low humidity, and high solar radiation, which have a direct and vital impact on human cognitive ability, especially the attention process [1,2,3,4]. Attention is usually divided into three functional components: alertness, selective attention, and sustained attention [5]. Sustained attention is defined as the ability to maintain an efficient level of detection and responsiveness to certain changes over a period of time [6,7,8]. It is dependent on the proper functioning of executive control mechanisms, particularly inhibitory control to regulate thought and action to conform with internal goals [9,10]. Inhibitory control, as the core of high-level cognitive functions, is a general control mechanism that includes flexibly monitoring the cognitive system [11], suppressing dominant or competitive responses [12,13], and optimizing the implementation of specific objectives [14,15]. Appropriate performance in inhibition-control tasks requires increased attentional resources allocated to task demands [16]. Previous studies have shown that sustained attention is influenced by Alzheimer’s disease [17], epilepsy [18], attention deficit hyperactivity disorder [19,20], and aging [21]. So far, it is not clear how the plateau environment affects sustained attention through inhibitory control mechanisms. With the increasing number of immigrants to the Qinghai–Tibet Plateau [22], it is increasingly important to focus on the effects of a low-oxygen environment on the cognitive function of lowlanders.

Previous studies have found that long-term chronic exposure to high altitude (HA) induces oxidative stress [23]. This stress exerts diverse effects on neurotransmitter levels (i.e., the synthesis of glutamate) in the central nervous system [24] and influences the normal and abnormal processes of sustained cognition by activating glutamatergic neurotransmission in the PFC [25], particularly inhibitory control processing in the executive function [24]. Among studies of chronic altitude hypoxia, the classic Go/No-Go task, which is relevant to the inhibition of dominant approach responses, has been commonly used to examine abnormal inhibitory control processing in populations with chronic hypoxia [26,27,28,29]. However, this paradigm is insufficient for evaluating the ability to maintain sustained attention over extended periods. In contrast, in the Go/No-Go sustained attention task (GSAT), participants were asked to respond to a high-proportion of Go stimuli and to withhold responses to the low proportion of No-Go target stimuli during three extended attention periods. Inhibitory control is critically in two aspects of this task: across Go trials to maintain task goals (i.e., withhold response when a low-frequency No-Go stimulus occurs), and after the reaction, especially following the No-Go trials where the subject failed to suppress the reaction (i.e., after an error) [30]. The GSAT is not only associated with top-down inhibitory control, but also explicitly reflects the distribution of attention resources over different periods of time. Higher omission errors during this task are associated with high-altitude hypoxia, related to the decline in the efficiency of cognitive control mechanisms [31]. Indeed, evidence from aging studies using event-related potentials (ERPs) has shown age-related differences in the recruitment of prepotent response inhibition over the course of the GSAT [21,32], in which the sustained potential (SP) following the stimulus, that is, maintaining attention across Go trials and withholding responses in No-Go trials, were related to inhibitory control [33]. In addition, a previous study used the classical Go/No-Go paradigm with the time-frequency approach to explore the neural oscillation courses of response inhibition, and found that the power of the delta (1–4 Hz) and theta (4–8 Hz) bands in the high-altitude area was lower than in the bands in the low-altitude area [34]. Roxburgh et al. (2020) also demonstrated that event-related oscillation (ERO) changes in the beta band (14–30 Hz) are relevant to frontal inhibitory control processing [35]. Heretofore, both time-frequency and time-domain analyses have rarely assessed how inhibition control is influenced by a high-altitude hypoxic environment sustained over time. However, evidence points to the fact that the application of ERPs (i.e., SP) and EROs markers (i.e., delta and theta) provides a new way to predict the performance of inhibitory control under sustained attention after long-term exposure to a high-altitude plateau environment.

So far, EEG studies on altitude hypoxia have mostly applied traditional statistical inference to compare between-group differences of high- and low-altitude populations, which would be exposed to type I or type II errors [36,37]. Additionally, the accuracy of relevant ERPs and neural oscillation signal characteristics for detecting related chronic hypoxia stress could not be confirmed. Machine learning is an approach that can solve type I and type II errors and improve reproducibility by reducing overfitting. This method is favored for its ability to analyze large and complex datasets and for its predictive accuracy when multiple predictive variables have complex interactions, through training, accumulating experience, and updating algorithms [38]. Automatic classification of machine learning is an important step toward making EEG more practical and less dependent on trained professionals for applications. At the most basic level, EEG datasets consist of 2D (time and channel) matrices of real values representing brain-generated potentials recorded on the scalp in relation to specific task conditions [39]. A great number of traditional machine learning algorithms such as logistic regression [40], SVM [41], decision tree [42], and random forest [43] have been applied on EEG data. In the clinical setting, EEG signals combined with machine learning have recently been used for identifying sleep disorders, epilepsy, strokes, and other neurological disorders [44]. To date, we have found only a flight hypoxia detection system designed to classify normal from hypoxic instances using the transformed EEG data, which were processed on the naïve Bayes, decision tree, random forest, and neural network algorithms, with sensitivity and specificity ranging from 0.83~1.00 and 0.91~1.00, respectively [45]. From a neuro-cognitive perspective, Knudsen’s attentional model proposed that inhibition of task-irrelevant processes played a vital role in sustained attention tasks [46,47]. However, the classification performance of inhibitory-control-related ERP components and neural oscillations in the GSAT as a tool for the auxiliary diagnosis of chronic altitude hypoxia stress has not been studied.

Therefore, in the current study, the GSAT was employed to explore the inhibitory control processes underlying sustained attention in HAs. At the behavioral level, we predicted that HAs would have higher omission errors in the No-Go trials, especially during the later period of sustained attention. At the neurological level, we expected that HAs would demonstrate certain abnormal neurophysiological indicators, which might manifest as difficulty withholding the response to No-Go trials by enhancing SP amplitude and increasing latency. Consistent with previous studies [48], we identified concerns about band-specific neural activities that are involved in chronic high-altitude hypoxia. Additionally, we examined whether ERP components and neural oscillations contained valuable information for discriminating between HAs and LAs. Classification accuracy was compared for different featur-selection techniques and classifiers, in order to identify the best selection method and classifier combination for assessing important biomarkers to help determine vulnerability to high-altitude hypoxia stress.

## 2. Methods

### 2.1. Participants

A total of 35 college students (16 females, 19 males; 22.200 ± 1.699 years) in high-altitude areas (Lhasa: 3680 m) were recruited from the Tibet University campus for the present study. These participants lived in low-altitude areas (<500 m) until early adulthood and had never lived at high altitude. At the time of the study, they had been living in Lhasa city for three years. The exclusion criterion of the high-altitude group was having a high-altitude experience before attending college.

We also collected data from 32 matched healthy college students (15 females, 17 males; 21.933 ± 1.438 years) in low-altitude areas (Guangzhou city) who had never lived at high altitude, as the control group. The exclusion criterion of the low-altitude group was having an experience of visiting the plateau. All participants were right-handed, free of neurological and psychiatric diseases, and had normal or corrected-to-normal vision. The two groups of participants were matched based on personality as tested by the NEO five-factor inventory (NEO-FFI) [49], intelligence scores on the Raven’s standard progressive matrices (SPM) [50], and their scores for the Pittsburgh sleep quality index [51] (all *p* > 0.05). Informed consent was obtained from each participant before the experiment, which was conducted in accordance with the Declaration of Helsinki and was approved by the Ethics Committee of Tibet University (XZTU2021ZRG-05).

### 2.2. Experimental Procedure

Settled in a dimly lit and sound-attenuated room, all participants were required to take part in a Go/No-Go sustained attention task (see Figure 1), which involved “Go trials” (the word and its font color were congruent) and “No-Go trials” (incongruent No-Go trials: the word and its font color were incongruent; repeat No-Go trials: the word and its font color were congruent but appeared in two consecutive trials). The trials were presented against a gray background and appeared 0.25° above a white point (in the center of a computer screen) at a distance of 100 cm. Participants were asked to press button ‘J’ when the “Go trials” stimuli were presented. In contrast, no response was required from participants when the “No-Go trials” appeared. The experiment consisted of two parts: first was the practice section of 60 trials prior to the formal experiments; the other was the formal experiments that included nine blocks lasting 72 min except for the practice part, each block consisting of 222 trials (198 Go trials and 24 No-Go trials). All stimuli were randomly selected and presented for 600 ms followed by a gray screen with a white point lasting approximately 1500 ms (in other words, the inter-stimulus interval was 1500 ms). The E-prime software system (Version 1.1, Psychology Software Tools, Inc., Pittsburgh, PA, USA) was used to present the experimental stimuli and collect behavioral response data.

### 2.3. EEG Recording and Analysis

Electroencephalograms (EEGs) were recorded from a 64-channel Neuroscan SynAmps2 amplifier (10/20 system) using Ag/Agcl electrodes mounted on a Neuroscan 64-channel QuickCap (Compumedics USA, Charlotte, NC, USA). EEG and vertical and horizontal electrooculogram (EOG) data were sampled at a rate of 500 Hz using an online bandpass filter of 0.01 Hz to 100 Hz. The data were re-referenced offline to the average for a bilateral mastoid on two ears, with a high-pass filter of 0.1 Hz (48 dB/oct) and a low-pass filter of 30 Hz (48 dB/oct). The impedances of all the electrodes were less than 10 kΩ. EOG were recorded bipolarly in order to monitor ocular artifacts.

#### 2.3.1. Time Domain Analysis

Curry 7.0 software was employed to remove ocular artifacts and bad blocks from the EEG signal. The rejection criterion for ocular artifacts was a negative or positive change of more than 200 μv, while the bad blocks caused by body movements or muscle activity with a negative or positive change of more than 100 μv were rejected. The averaged waveforms for each participant were calculated for each station. The stimulus-locked data were segmented into epochs of 200 ms before and 1000 ms after stimulus onset [52,53], including the baseline-corrected interval of −200 ms to 0 ms. The stimuli-locked ERPs (SP) in the Go and No-Go trials were averaged. At the electrode site of the CPZ, the maximum frontal and parietal amplitudes of the SP were observed within the 460 to 700 ms time window.

#### 2.3.2. Time-Frequency Analysis

A time-frequency analysis was performed using the MATLAB toolbox FieldTrip [54], epoched into a period of −200 to 1000 ms, corresponding to the onset of the word stimulus. A short-time Fourier transform method was applied to estimate event-related spectral perturbation (ERSP). Spectral power was calculated on artifact-free epochs using a fast Fourier transform (FFT) with a Hanning window at a resolution of 0.1 Hz, averaged across all trials, and then converted to log power. Baseline-normalized ERSP was acquired by subtracting the average baseline log power spectrum from each spectral estimate (in dB). Again, since we aimed to compare the time-frequency responses to successful restraint and cancellation, the results for each participant of the successful Go and No-Go trials were separately averaged for further analysis. Based on the grand-averaged time-frequency plots in the present study and the existing literature using Go/No-Go and stop-signal tasks, we specifically examined the powers of theta [55,56], alpha [57,58], and beta oscillations [58,59,60]. The time window of interest was selected from the grand-averaged time-frequency plots. In the present study, the theta (4–8 Hz), slow alpha (8–10.5 Hz), fast alpha (10.5–13 Hz), and beta (13–30 Hz) powers were defined as the mean of the time window of 460 to 700 ms, respectively.

### 2.4. Statistical Analysis

#### 2.4.1. Analysis of Demographic Data

For the demographic data analysis, *t*-tests were performed on participants’ scores on the Pittsburgh sleep quality index (PSQI) [61], Raven’s standard progressive matrices (SPM) [62], and the neuroticism-extraversion-openness five-factor inventory (NEO-FFI) [63].

#### 2.4.2. Behavioral Data

The experiment was divided into three periods (a1, a2, a3), which correspond to the early, middle, and later stages of sustained attention. Each period included three task blocks lasting 24 min before the ERP and performance data were analyzed. At the behavioral level, the reaction times (RTs) for hits on the Go trials and the rate of omission errors (the miss rate in the Go trials) were recorded. Behavioral data were analyzed using repeated-measures ANOVA with group (HA vs. LAs) as the between-subjects factor and period (a1 vs. a2 vs. a3) as the within-subjects factor.

#### 2.4.3. Time-Domain and Time-Frequency Analysis

To evaluate the characteristics of response inhibition in HA, ERPs and ERSP were compared between HA and LA participants. In particular, time-domain analyses mainly focused on the SP components, whereas the time-frequency analysis mainly focused on the theta, alpha, and beta frequencies. Since the alpha frequency band contains more unique information, we divided this band into sub-slow alpha (8–10.5 Hz) and fast alpha (10.5–13 Hz) frequency bands.

The electrophysiological indicators (i.e., mean amplitude and latency of SP, the power intensity in delta, slow alpha, and fast alpha) were assessed by repeated-measures ANOVA with group as the between-subject factor and period and stimulus type (Go vs. No-Go) as within-subjects factors. Greenhouse-Geisser corrected *p* values were calculated when necessary. Post-hoc *t*-tests (two-tailed) were applied to examine significant interactions. Two-tailed tests at an a priori threshold of *p* < 0.05 were established to indicate statistical significance.

### 2.5. Classification Methodology

#### 2.5.1. Feature Generation

##### Time-Domain Features

Twelve time-domain features (at the 460–700 ms time interval) were selected for the SP components obtained at the CPZ electrodes in the Go and No-Go conditions.

(1) Latency, i.e., the time at which the maximum peak value of the SP component occurs:(1)$$t_{S_{max}^{SP}}^{SP}=\{t|S^{SP}\;\left(t\right)=S_{max}^{SP}\}$$

(2) Mean amplitude, which is the mean of the amplitude values for the SP component from 460 to 700 ms:(2)$$S_{mean}^{SP}=(\sum_{t=460}^{t=700}S^{SP}\;\left(t\right))/t$$

(3) Total area, which is the sum of the signal values for the SP component from 460 to 700 ms:(3)$$A_{S}^{MFN}=\sum_{t=460}^{t=700}S^{SP}\;\left(t\right)$$

(4) Latency/mean amplitude ratio:(4)$$t_{S_{max}^{SP}}^{SP}/S_{mean}^{SP}$$

(5) Absolute mean amplitude:(5)$$\left|S_{mean}^{SP}\right|$$

(6) Absolute latency/amplitude ratio:(6)$$\left|t_{S_{max}^{SP}}^{SP}/S_{mean}^{SP}\right|$$
where $t^{sp}$ is the time window corresponding to the SP and $S^{SP}\;\left(t\right)$ is the signal value corresponding to the SP.

##### Time-Frequency Domain Features

The short-time Fourier transform was used to convert the time-domain signal into a time-frequency signal.
(7)$$STFT\left(t,f\right)=\int_{-\infty }^{\infty }x\left(\tau \right)h\left(\tau -t\right)e^{-j2\pi fz}d\tau$$

The time-frequency features were selected from the CPZ electrodes. Four frequency bands consisting of theta, slow alpha, fast alpha, and beta bands were extracted under Go and No-Go conditions, respectively. The time intervals corresponded to the time windows of the SP (460–700 ms). The mean power, maximum power value, and time points corresponding to the maximum power value were calculated for the above time periods of the corresponding frequency bands. Each participant had 24 features (two stimulus conditions × three time-frequency feature values × four frequency bands).

(1) The mean power in the above time periods (460–700 ms) was calculated to obtain the average power values of the corresponding frequency bands:(8)$$f_{mean}^{SP}=(\overset{t=700}{\underset{t=460}{\sum }}\frac{\sum_{f=-\infty }^{f=\infty }STFT\left(t,f\right)}{f})/t$$

(2) The maximum power value was calculated using two steps: Firstly, the power values were averaged at each time point in different frequency bands from 460–700 ms. We apply the maximum of these power averages:(9)$$f_{max}^{SP}=\max((\overset{f=\infty }{\underset{f=-\infty }{\sum }}STFT\left(t,f\right))/f)$$

(3) The time points corresponding to the maximum average power value:(10)$$t_{f_{max}^{SP}}^{SP}=\{t|(\overset{f=\infty }{\underset{f=-\infty }{\sum }}STFT\left(t,f\right))/f=f_{max}^{SP}\}$$
where *STFT*(*t, f*) is the result of the short-time Fourier transform, where *t* stands for time and *f* stands for frequency.

#### 2.5.2. Feature Selection

As a data preprocess in machine learning, feature selection plays an important role in dimensionality reduction and irrelevant data removal to improve classification accuracy. There were 12 time-domain and 24 frequency-domain features in this experiment.

First, the feature data were processed as dimensionless because of the large difference in the data values between the selected features, which may affect the data classification result. In this study, the min–max normalization method was used for processing the dimensionless data. This method maps the original data to between [0, 1] through a linear transformation, and the formula is as follows:(11)$$X^{\prime }=\frac{X-X_{min}}{X_{max}-X_{min}}$$
where *X*′ represents the final result and *X* represents the original data, and $X_{min}$ and $X_{max}$ are the minimum and maximum values of the data distribution, respectively.

Second, feature selection was followed by min–max normalization. A higher feature dimension influences the results of the classifier to a certain extent and reduces its efficiency. Therefore, we adopted two feature selection methods, ReliefF and GainRatio (GR). The intersection of the results of the two feature selection methods was then considered the final feature.

ReliefF, first proposed by Robnikšikonja and Kononenko [64], is a feature-sorting method based on the distance between features. The basic idea is to select randomly a sample from the training set and search for K-nearest neighbor samples from the same and different categories. The feature weights were updated according to the formula, and the relevant features were selected according to the weights.

The GainRatio (GR) feature-selection method calculates the information-gain rate of each feature and selects features according to information gain. Because information gain is greater for features with more values, the information gain rate overcomes some disadvantages; therefore, the GR method was adopted. *SplitInfoF* (*S*) was applied to reduce the influence of information gain. The formula used is as follows:(12)$$SplitInfoF\;\left(S\right)=-\sum_{i=1}^{v}\frac{\left|S_{v}\right|}{\left|S\right|}\times log_{2}\left(\frac{\left|S_{v}\right|}{\left|S\right|}\right)$$

$S_{v}$ represents the set of class *v* data. The calculation formula for the information gain rate is as follows:(13)$$GainRatio\;\left(F,S\right)=\frac{Gain\;\left(F\right)}{SplitInfoF\;\left(S\right)}$$

*Gain* (*F*) represents information gain. The above two feature selection methods—ReliefF and GR—were used for feature selection in the time and frequency domains, respectively. Each classifier extracted 36 features, including 12 time-domain and 24 frequency-domain features. All features generated by each classifier consisted of two permutations and were ranked from high to low. The first 18 features of the two permutations were selected as two datasets, and the intersection of the two datasets was utilized to obtain eight time-domain and three frequency-domain features.

#### 2.5.3. Classifiers

As there is no unified classifier for this type of research, we used a variety of classifiers to test and compare the results of each classifier. In this study, we chose a support vector machine (SVM), logistic regression (LR), and a decision tree (DT) to construct the classification model. The classification accuracy of each classifier for each component was tested using the leave-one-out cross-validation (LOOCV) method. This method keeps one sample at a time for the test set and the other samples for the training set. If there are k samples, it is necessary to train k times and test k times. The LOOCV calculation is complicated, but its sample utilization rate is high. Therefore, it is suitable for use with small sample sizes.

An SVM can find hyperplane partition data in a given sample space to achieve data classification. It is also possible to classify nonlinear data by mapping the data to a higher-dimensional space through kernel functions.

Logistic regression is a dichotomous technique, and its mathematical model is expressed as follows:(14)$$h_{\theta }\;\left(x\right)=g\;\left(\theta^{T}x\right)$$
(15)$$g\;\left(z\right)=\frac{1}{1+e^{-z}}$$
*x* represents each feature of the training set, *θ* is the parameter of the model, and $g\;\left(z\right)$ is the activation function, which is an s-shaped curve. The results obtained by $\theta^{T}x$ are mapped to [0, 1].

A decision tree is a tree-like process classification algorithm, which creates a number of internal nodes and leaf nodes to make a series of “yes” or “no” judgments, thereby achieving data classification. The leaf node represents the final classification result, and the branch path represents the classification rule. In addition, there are attribute selection methods that usually include the Gini index and information gain. The Gini index was also used in this study.

#### 2.5.4. ROC Analysis

The ROC curve was computed from a set of dichotomous training data used for learning, pre-supplied with the data’s categorical labels. Sensitivity and specificity at different data thresholds were calculated by analyzing the measurement results from a dichotomous population. The area under the curve (AUC) is generally used as an index to evaluate the accuracy of ROC analysis, reflecting the accuracy of classification. In our study, the AUC value was used as the evaluation criterion to indicate clearly and accurately which classifier had a better classification effect. In other words, the AUC value was proportional to the classification effect of the classifier (Table 1).
(16)$$\mathrm{Accuracy}=\frac{TP+TN}{TP+TN+FP+FN}$$
(17)$$F1\text{-}\mathrm{score}=2\times \left(\frac{precision\times recall}{precision+recall}\right)$$
(18)$$\mathrm{Sensitivity}=\frac{TP}{TP+FN}$$
(19)$$\mathrm{Specificity}=\frac{TN}{TN+FP}$$

AUC: The area under the curve.

## 3. Results

### 3.1. Demography Data

Independent *t* testing and Chi-square tests were performed on age, gender, PSQI, SPM, and NEO-FFI to establish whether the HA and LA groups matched each other regarding demographic data (Table 2).

### 3.2. Behavior Performance

Regarding the omission error rate, two-way mixed-design ANOVAs revealed that there was a significant main effect from altitude [*F*(1, 65) = 24.01, *p* < 0.001, $\eta_{p}^{2}$ = 0.270], and the omission error rate (1.33 ± 1.04%) for the high-altitude group was smaller than that of the low-altitude group (3.98 ± 3.01%). A significant main effect for period was observed [*F*(2, 64) = 6.23, *p* = 0.003, $\eta_{p}^{2}$ = 0.163], and the omission errors in the a1 period (2.82 ± 2.69%) were larger than those in the a2 period (2.20 ± 2.31%). A marginally significant interaction between altitude and period was observed [*F*(2, 64) = 3.12, *p* = 0.051, $\eta_{p}^{2}$ = 0.089]. In the a1 period, the high-altitude group (1.54 ± 1.26%) made fewer omission errors than the low-altitude group (4.21 ± 3.13%) [*F*(1, 65) = 21.64, *p* < 0.001, $\eta_{p}^{2}$ = 0.250], and the same trend was observed in the a2 period [*F*(1, 65) = 13.90, *p* < 0.001, $\eta_{p}^{2}$ = 0.176] and a3 period [*F*(1, 65) = 11.89, *p* = 0.001, $\eta_{p}^{2}$ = 0.210]. Pairwise comparisons showed that the low-altitude group made more omission errors during period a1 than during a2 (*p* = 0.036), whereas in the high-altitude group, omission errors remained stable with time spent on the task [*F*(2, 33) = 3.02, *p* = 0.062, $\eta_{p}^{2}$ = 0.155] (Figure 2).

The mean reaction times in the Go trials revealed a significant effect of period [*F*(2, 64) = 11.87, *p* < 0.001, $\eta_{p}^{2}$ = 0.271]. Reaction times significantly decreased over the course of the task, reaction times in period a1 (461.58 ± 88.90) were larger than those in periods a2 (431.49 ± 83.23) and a3 (426.53 ± 76.22). Furthermore, the values in period a2 (431.49 ± 83.23) were larger than in period a3. No significant main effect of altitude [*F*(1, 65) = 0.35, *p* = 0.557, $\eta_{p}^{2}$ = 0.005] or interaction between altitude and period [*F*(2, 64) = 0.78, *p* = 0.461, $\eta_{p}^{2}$ = 0.024] were found (Table 3).

### 3.3. Time-Domain Analysis

Regarding SP latencies, a significant main effect of stimulus type was observed [*F*(1, 65) = 143.21, *p* < 0.001, $\eta_{p}^{2}$ = 0.688]. The latencies in Go trials (517.82 ± 5.81) were shorter than in No-Go trials (583.78 ± 5.14). In addition, effects of altitude [*F*(1, 65) = 14.50, *p* = 0.001, $\eta_{p}^{2}$ = 0.182] and the period × altitude interaction effect [*F*(2, 130) = 3.48, *p* < 0.05, $\eta_{p}^{2}$ = 0.051] were observed. For the above two-way interaction, follow-up simple effects testing revealed that HA participants had significantly shorter SP latency than LAs for periods a1 (HA: 531.51 ± 8.05, LA: 579.63 ± 7.60; *p* < 0.001) and a3 (HA: 528.14 ± 8.00, LA: 565.97 ± 7.05; *p* < 0.01), while no significant effect by group was observed in period a2 (HA: 538.57 ± 7.27, LA: 560.97 ± 8.69; *p* > 0.05) (Table 4).

For SP amplitude, the main effects were stimulus type [*F*(1, 65) = 287.07, *p* < 0.001, $\eta_{p}^{2}$ = 0.815] and period [*F*(2, 130) = 5.81, *p* < 0.01, $\eta_{p}^{2}$ = 0.082], as well as stimulus type × period interaction [*F*(2, 130) = 5.02, *p* < 0.01, $\eta_{p}^{2}$ = 0.072]. Follow-up simple effects analysis revealed that both groups had greater SP amplitude for No-Go trials during periods a2 (13.61 ± 0.66) and a3 (14.06 ± 0.69) than period a1 (12.42 ± 0.65; *ps* < 0.05) for the CPz channels, but not for the Go trials [*F*(2, 132) = 0.87, *p* > 0.05, $\eta_{p}^{2}$ = 0.013] (Table 5, Figure 3 and Figure 4).

### 3.4. Time-Frequency Analysis

#### 3.4.1. Theta

The ANOVA testing performed on the theta power results revealed a significant interaction between altitude and period (*F*(2, 64) = 5.17, *p* < 0.01, $\eta_{p}^{2}$ = 0.139). Comparisons indicated that the theta power of HAs was significantly higher than LAs at periods a2 (HA: 0.52 ± 0.11, LA: 0.24 ± 0.08; *p* < 0.05) and a3 (HA: 0.78 ± 0.16; LA: 0.22 ± 0.09; *p* < 0.01). In HAs, theta power increased significantly between periods a1 and a3 (a1: 0.42 ± 0.09; a3: 0.78 ± 0.16; *p* < 0.05), while remaining stable with time spent on task in LAs (Table 6 and Figure 5).

#### 3.4.2. Slow Alpha

Regarding slow alpha power, ANOVA revealed the significant main effect of stimulus type [*F*(1, 65) = 207.71, *p* < 0.001, $\eta_{p}^{2}$ = 0.762]. Meanwhile, a significant effect from period × stimulus type × altitude interaction [*F*(2, 130) = 3.39, *p* < 0.05, $\eta_{p}^{2}$ = 0.050] was observed: the LA group showed relatively lower slow alpha band activation in the No-Go trials compared to the HA group in the a2 period (*t* = −2.28, *p* < 0.05), whereas similar trends were found in both the Go and No-Go trials in period a3 (*t* = −3.98, *p* < 0.001; *t* = −3.21, *p* < 0.01) (Table 6 and Figure 5).

#### 3.4.3. Fast Alpha

For fast alpha power, the main effects of stimulus type and interaction effect of period × stimulus type were significant [*F*(1, 65) = 12.03, *p* < 0.01, $\eta_{p}^{2}$ = 0.156; *F*(1, 65) = 6.18, *p* < 0.01, $\eta_{p}^{2}$ = 0.162]. Follow-up simple effects analysis revealed that fast alpha power in period a1 was larger than in period a3 (a1: −0.12 ± 0.03, a3: −0.18 ± 0.04; *p* < 0.05). No significant main effects of altitude [*F*(1, 65) = 1.23, *p* = 0.271, $\eta_{p}^{2}$ = 0.019] or interaction between altitude and period [*F*(2, 130) = 1.27, *p* = 0.283, $\eta_{p}^{2}$ = 0.019] were found (Table 6 and Figure 5).

#### 3.4.4. Beta

For beta power, ANOVA revealed the significant interaction effect of period × stimulus type × altitude interaction [*F*(2, 130) = 6.77, *p* < 0.01, $\eta_{p}^{2}$ = 0.094]. Follow-up simple effects analysis revealed that HAs had greater beta activities in the No-Go trials than LAs in period a1 (*t* = 2.62, *p* < 0.05). In addition, the same trend was observed in both Go and No-Go trials in periods a2 (*t* = 2.20, *p* < 0.05; *t* = 2.71, *p* < 0.01) and a3 (*t* = 4.09, *p* < 0.001; *t* = 2.64, *p* = 0.010) (Table 6 and Figure 5).

### 3.5. Machine Learning

Figure 6 shows the distribution of extracted features by violin plots. Meanwhile, Table 6 shows the SVM, LR, and DT classifiers’ evaluation criteria, including precision, sensitivity, specificity, F1 score, and AUC for the ERP and ERSP datasets. The performance achieved by SVM increased gradually across all three periods and reached the highest accuracy of 92.54% in period a3 (Table 7). The classification accuracy of LR peaked at 95.52% in period a1 and then decreased over time (Figure 7 and Figure 8). Among the three different machine learning algorithms, the classification performance of DT was the worst, with accuracies for all three time periods falling below 85.00%.

## 4. Discussion

The present study assessed response inhibition in HA participants in a situation of sustained attention, by examining their oscillatory components within the time and frequency domains in a sustained attention Go/No-Go task. To determine a more efficient method for detecting HA, three classification algorithms—SVM, LR, and DT—were evaluated. The following conclusions were drawn from the results.

Our behavioral performance measure to track response inhibition in sustained attention was the omission error rate, which is a widely used assessment of attentional vigilance and is considered sensitive to a variety of performance results, such as hypoxia [65], aging [66], and sleep deprivation [67]. We found that the HA group had a lower omission error rate than the LA group in sustained attention performance across all periods, suggesting that HA participants exhibited increased vigilance and a lower level of response inhibition. Similarly, our previous findings suggested that chronic exposure to high altitudes resulted in overactive performance monitoring [68]. Therefore, it might be speculated that chronic hypoxic stress provokes a compensatory mechanism, resulting in adaptation to long-term exposure to hypoxic environments.

Previous studies by our team have shown that passively induced ERPs are disrupted during chronic high-altitude hypoxic exposure [69,70]. This study is the first to demonstrate the utility of ERPs in tracking the time course of recovery following hypoxia. Our results demonstrated that HA participants had relatively shorter latency of the SP component over the medial frontal regions in periods a1 and a3 compared with LA individuals. Studies using electrophysiological recording have noted that SP is a marker of conflict resolution and response selection [71,72]. Therefore, our results suggest that the latency of the SP was shorter in the HA group than in the LA group, perhaps because of increasing vigilance in their responses to conflict resolution and enhanced adaptation to conflict. This is consistent with the results of previous studies showing that cognitive decrements occurred only when exposed to high levels of hypoxia [73], while exposure to mild altitude levels (3000 and 4000 m) did not degrade executive function (i.e., Stroop color test performance) [74].

We conducted quantitative analyses of the EEG signal rhythms by estimating the power intensity of the frequency bands. Midline frontal theta activity is associated with response inhibition [75]. Stronger theta activities in the prefrontal regions during the middle and later periods of sustained attention were observed in HAs compared with LAs. In addition, theta band power was significantly enhanced over the three periods in the HA group but not in the LA group. These findings could be a possible explanation for compensatory cognitive scaffolding in high-altitude immigrants, accompanied by an increase in functional engagement in the frontal regions. Second, alpha power is supposed to be associated with relatively higher internal attention demands [76] and with the suppression of irrelevant information, allowing the maintenance of attention during task execution [77]. Slow alpha (8~10.5 Hz) and fast alpha (10.5~13 Hz) oscillations have been reported to be modulated by top-down preparations for response control [77] and sensory-semantic processes, respectively [78,79]. The slow alpha band oscillation results showed that the HA group invested more attentional resources in detecting conflict information than the LA group, especially during the middle and later stages of sustained attention. Although there are differences between slow and fast alpha activities, it is worth noting that alpha activity was found to be the most sensitive neuroelectrophysiological indicator that could be used in detecting fatigue [80]. These results suggest that plateau immigrants are more likely to experience cumulative fatigue. Interestingly, the fast alpha rhythms that were associated with the sensory-semantic processes during inhibition (No-Go) [75] were found to not be influenced by high-altitude-induced chronic stress. Third, previous studies have shown that increased prefrontal beta oscillations indicate enhanced cognitive control [81,82]. The current study implies that increased adaptation to conflict over sustained attention periods was accompanied among HAs by a greater need for cognitive resources in the conflict-processing stage. In accordance with the compensation hypothesis of neural circuit utilization [83,84], the PFC cortical areas of lowlanders exposed to high altitude may have been more strongly activated during the GSAT, in order to compensate for decreased oxygen availability on the plain.

To our knowledge, this study is the first to investigate the classification accuracy of ERPs and EROs underlying sustained attention processes using machine learning. We found that the classification performance of SVM was superior to the other two algorithms (LR and DT), and reached an accuracy of 92.54% during the last period of the sustained attention process. The F1 score, sensitivity, specificity, and AUC of SVM also achieved optimal performance during this period, which was consistent with previous studies, indicating that SVM machine learning can effectively distinguish human fatigue under high-altitude, low-oxygen conditions compared with control subjects [85]. Compared to previous studies [55], we achieved good classification performance by using 11 features from one electrode. The current findings indicate that the SP induced by “Go” and “No-Go” trials and the neural oscillations corresponding to the time window contain more sensitive information and may be helpful for assisting in the identification of oxidative stress, which is associated with chronic exposure to a high-altitude environment.

Above all, the present findings support the notion that HA participants display compensatory activation of cognitive control resource consumption during sustained attention processing. However, because of the relatively small sample size of acute hypobaric hypoxia, we could not compare the different characteristics of response inhibition in oxygen deficiency in migrants to plateaus when dealing with overcoming habitual responses. This was a limitation of the present study. We are continuing to collect data on the high-altitude hypoxic population described above. In addition, because of the high trial-to-trial variability and the unfavorable ratio between signal (ERP/EROs) and noise, implementation classification using ERP and EROs on a single-trial basis and extracting effective features for real-time identification of hypoxia stress at high altitude are also challenging problems. In future studies, effective feature-extraction algorithms should be developed to identify chronic high-altitude hypoxia.

We should consider several issues in future work. First, the inter-trial phase locking of neural oscillations characteristic of hypoxia tolerance is a measure of phase consistency of the neural activities, and experimental trails would be a potential way to better understand the neural mechanisms of human adaptation to high altitudes [86]. Second, because the amplitude of the EEG signal is particularly sensitive to noise, a multiple linear regression model of the EEG power spectrum should be explored in relation to the role of sustained attention in chronic high-altitude hypoxia. Third, deep learning should be combined with machine learning, which may bring a key step toward helping psychologists to classify accurately and reliably chronic high-altitude hypoxia and plain controls.

## 5. Conclusions

Our work found that in HA individuals the PFC areas specific to cognitive tasks were activated to compensate for reduced oxygen availability compared to subjects from low-altitude areas, which was reflected in the time-domain (e.g., SP) and time-frequency-domain (e.g., delta, slow alpha, fast alpha, and beta) datasets. Furthermore, data mining indicated that SVM is a more effective classification method to discriminate between chronic hypoxic exposure and a normoxic baseline at a later stage of sustained attention, compared with the LR and DT algorithms. These findings not only provide novel insights into response inhibition in a situation of sustained attention in chronic exposure to high-altitude environments, but also reveal an auxiliary method for identifying chronic plateau hypoxia at the electroneurophysiological level.

## Figures and Tables

**Figure 1 brainsci-12-01677-f001:**
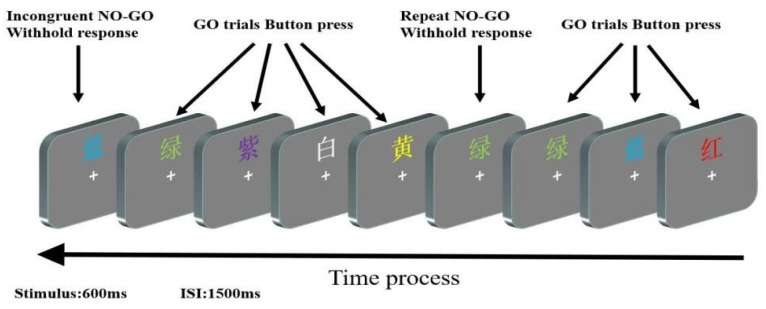
The procedure of the experimental paradigm. “蓝” = Blue; “绿” = Green; “紫” = Purple; “白” = White; “黄” = Yellow; “红” = Red.

**Figure 2 brainsci-12-01677-f002:**
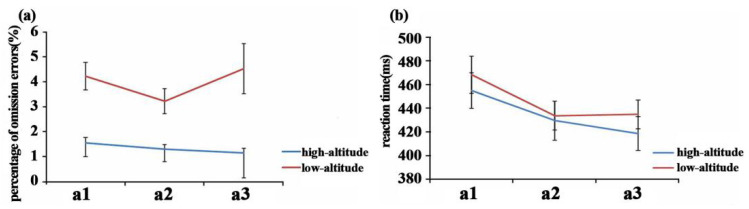
(**a**) The percentage of omission errors and (**b**) the mean of the reaction time on the two groups for the periods (a1/a2/a3).

**Figure 3 brainsci-12-01677-f003:**
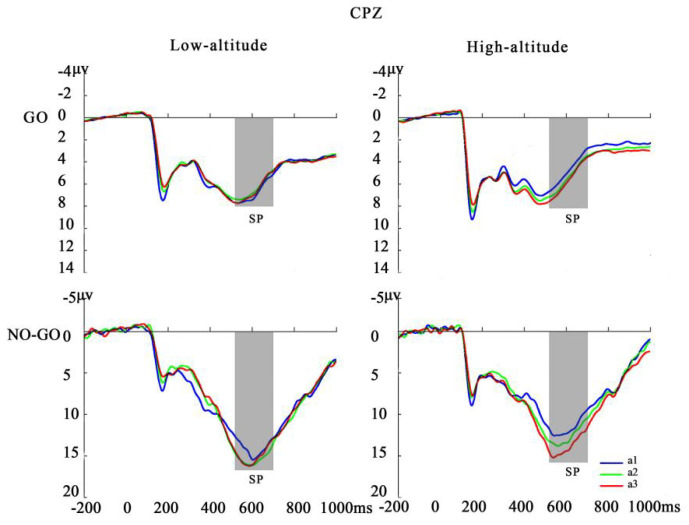
Stimulus-locked ERP waveforms as a function of altitude (low-altitude/high-altitude), period (a1/a2/a3), stimulus type (Go/No-Go), and electrode site (CPZ).

**Figure 4 brainsci-12-01677-f004:**
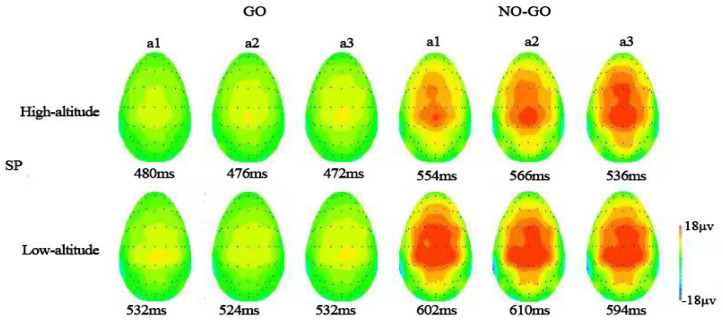
The images represent activity on the scalp at the time corresponding to the maximum amplitude of each component for each group.

**Figure 5 brainsci-12-01677-f005:**
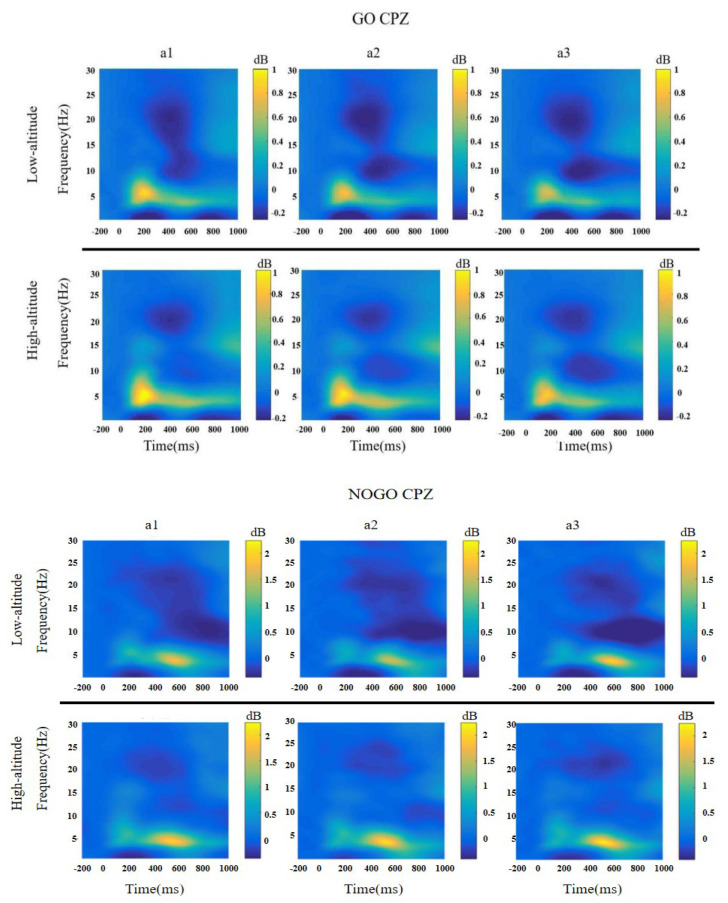
The average results of spectro-temporal analysis of EEG activity recorded from CPZ electrode sites during Go and No-Go trials.

**Figure 6 brainsci-12-01677-f006:**
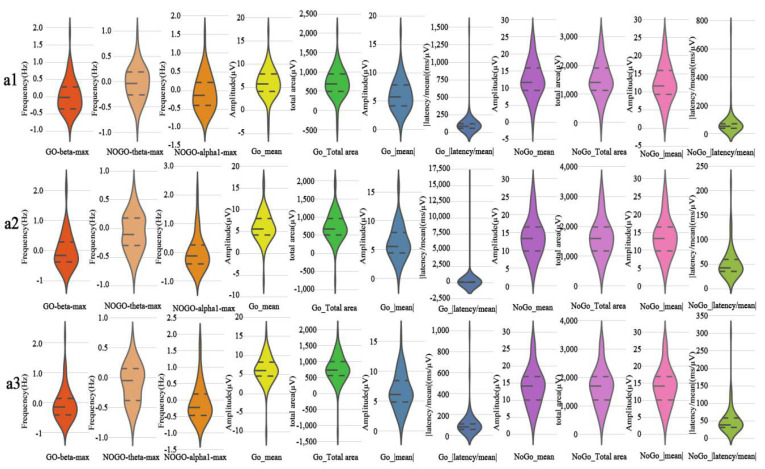
Violin diagrams showed the distribution of extracted features. GO-beta-max = Maximum power value of beta band during Go trials; NoGo-theta-max = maximum power value of theta band during No-Go trials; NOGO-alpha1-max = maximum power value of alpha1 band during No-Go trials; Go_mean = average amplitude of SP components during Go trials; Go_Total area = total area of SP components during Go trials; Go_|mean| = absolute value of the mean amplitude of SP component during Go trials; Go_|latency/mean| = absolute value of the ratio between latency and mean amplitude of SP components during Go trials; NoGo_mean = average amplitude of SP components during No-Go trials; NoGo_Total area = total area of SP components during No-Go trials; NoGo_|mean| = absolute value of the mean amplitude of SP component during No-Go trials; NoGo_|latency/mean| = absolute value of the ratio between latency and mean amplitude of SP components during No-Go trials.

**Figure 7 brainsci-12-01677-f007:**
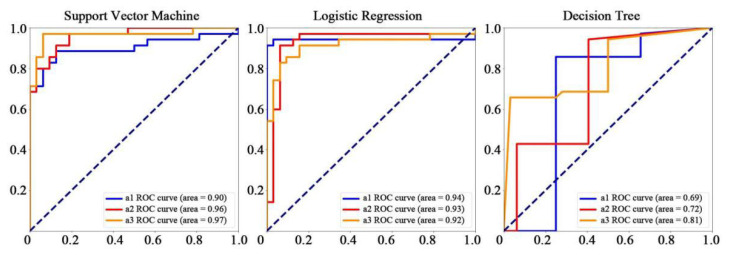
ROC curves of the three classifiers—SVM, LR, and DT—based on the ERP (i.e., SP) and ERSP (i.e., delta, slow alpha, fast alpha, and beta frequency bands) datasets at the CPz electrode across a1, a2, and a3 periods, respectively.

**Figure 8 brainsci-12-01677-f008:**
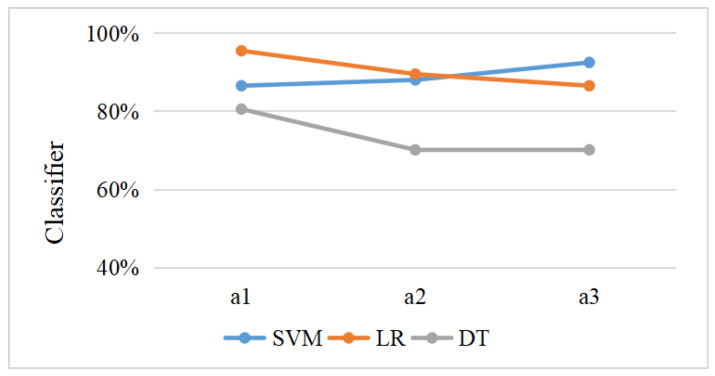
Changes in the accuracy of machine learning algorithms SVM, LR and DT over the course of periods a1, a2, and a3.

**Table 1 brainsci-12-01677-t001:** The metrics used for evaluating performance of classifiers.

		True Value	
		Positive	Negative
Predictive value	Positive	*TP*	*FN*
	Negative	*FP*	*TN*

Note: *TP* = True Positive; *FN* = False Negative; *FP* = False Positive; *TN* = True Negative.

**Table 2 brainsci-12-01677-t002:** Demographic data in high-altitude and low-altitude subjects.

	High-Altitude (M ± SD)	Low-Altitude (M ± SD)	*t*	*p*
Age (years)	22.200 ± 1.699	21.933 ± 1.438	0.691	0.492
Gender (male/female)	19/16	17/15	0.009	0.924
PSQI	3.400 ± 2.291	3.281 ± 2.098	0.221	0.826
SPM	44.686 ± 15.976	47.250 ± 9.374	−0.792	0.431
NEO-FFI	177.286 ± 56.098	194.469 ± 11.086	−1.775	0.084

Note: PSOI = Pittsburgh sleep quality index; SPM = Raven’s standard progressive matrices; NEO-FFI = neuroticism–extraversion–openness.

**Table 3 brainsci-12-01677-t003:** The mean RTs in high- and low-altitude subjects (mean ± SD).

Stimulus	Period	HA Group (n = 35)	LA Group (n = 32)
Go	a1	491.43 ± 10.26	540.56 ± 10.69
	a2	503.03 ± 9.48	537.69 ± 9.51
	a3	491.94 ± 8.23	542.25 ± 8.44
No-Go	a1	571.60 ± 8.90	618.69 ± 8.54
	a2	574.11 ± 9.82	584.25 ± 10.77
	a3	564.34 ± 11.13	589.69 ± 9.90

**Table 4 brainsci-12-01677-t004:** Mean latencies of the SP (ms) (M ± SEM).

Stimulus	Period	HA Group (n = 35)	LA Group (n = 32)
Go	a1	5.02 ± 0.58	6.89 ± 0.45
	a2	5.60 ± 0.61	6.56 ± 0.55
	a3	5.84 ± 0.69	6.69 ± 0.55
No-Go	a1	11.39 ± 0.93	13.55 ± 0.87
	a2	12.52 ± 0.82	14.80 ± 1.04
	a3	13.62 ± 1.00	14.54 ± 0.96

**Table 5 brainsci-12-01677-t005:** Mean amplitudes of the SP (μV) (M ± SEM).

Stimulus	Period	HA Group (n = 35)	LA Group (n = 32)
Go	a1	5.02 ± 0.58	6.89 ± 0.45
	a2	5.60 ± 0.61	6.56 ± 0.55
	a3	5.84 ± 0.69	6.69 ± 0.55
No-Go	a1	11.39 ± 0.93	13.55 ± 0.87
	a2	12.52 ± 0.82	14.80 ± 1.04
	a3	13.62 ± 1.00	14.54 ± 0.96

**Table 6 brainsci-12-01677-t006:** Mean ERSP values for delta, slow alpha, fast alpha, and beta frequency bands (dB) (460–700 ms) (M ± SEM).

	Stimulus	Period	HA Group (n = 35)	LA Group (n = 32)
theta	Go	a1	0.84 ± 0.15	0.84 ± 0.16
		a2	1.07 ± 0.16	0.74 ± 0.16
		a3	1.44 ± 0.24	0.88 ± 0.25
	No-Go	a1	−0.01 ± 0.07	−0.19 ± 0.08
		a2	−0.03 ± 0.09	−0.27 ± 0.09
		a3	0.13 ± 0.09	−0.43 ± 0.09
slow alpha	Go	a1	−0.10 ± 0.05	−0.14 ± 0.05
		a2	−0.13 ± 0.05	−0.17 ± 0.05
		a3	−0.17 ± 0.06	−0.20 ± 0.05
	No-Go	a1	−0.01 ± 0.08	−0.18 ± 0.07
		a2	−0.02 ± 0.11	−0.27 ± 0.07
		a3	−0.02 ± 0.10	−0.42 ± 0.04
fast alpha	Go	a1	−0.09 ± 0.05	−0.17 ± 0.06
		a2	−0.12 ± 0.05	−0.18 ± 0.06
		a3	−0.15 ± 0.06	−0.18 ± 0.06
	No-Go	a1	−0.14 ± 0.07	−0.28 ± 0.07
		a2	−0.03 ± 0.09	−0.31 ± 0.09
		a3	−0.11 ± 0.08	−0.37 ± 0.08
beta	Go	a1	−0.04 ± 0.02	−0.11 ± 0.02
		a2	−0.03 ± 0.02	−0.10 ± 0.02
		a3	−0.05 ± 0.02	−0.08 ± 0.02
	No-Go	a1	−0.07 ± 0.03	−0.17 ± 0.03
		a2	−0.05 ± 0.04	−0.19 ± 0.04
		a3	−0.09 ± 0.03	−0.16 ± 0.03

**Table 7 brainsci-12-01677-t007:** Precision, sensitivity, and specificity values achieved for ERPs and EROs by the SVM, LR, and DT algorithms for the Elman neural network.

Classifiers	Periods	Accuracy (95% Confidence Interval)	F1 Score	Sensitivity	Specificity	AUC
SVM	a1	86.57% (78.41%, 94.73%)	0.89	85.71%	87.50%	0.90
	a2	88.06% (80.3%, 95.82%)	0.88	91.43%	84.38%	0.96
	a3	92.54% (86.25%, 98.83%)	0.93	91.43%	93.75%	0.97
LR	a1	95.52% (90.57%, 100.00%)	0.96	91.43%	100.00%	0.94
	a2	89.55% (82.22%, 96.88%)	0.90	91.43%	87.50%	0.93
	a3	86.57% (78.41%, 94.73%)	0.87	88.57%	84.38%	0.92
DT	a1	80.60% (71.13%, 90.07%)	0.80	85.71%	75.00%	0.69
	a2	70.15% (59.19%, 81.11%)	0.68	80.00%	59.38%	0.70
	a3	70.15% (59.19%, 81.11%)	0.70	65.71%	75.00%	0.81

## Data Availability

Data materials can be obtained by contacting the corresponding author.

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
