# Peer review of "Machine Learning Based on Event-Related EEG of Sustained Attention Differentiates Adults with Chronic High-Altitude Exposure from Healthy Controls"

_brainsci, 2022, doi:10.3390/brainsci12121677_

Round 1
Reviewer 1 Report
The paper aims to examine the effect of high altitude on inhibitory control processes that underlie sustained attention in the neural correlates of EEG data. Machine learning approaches involving EEG data of inhibitory control were used to classify high altitude chronic hypoxia and plain controls. The paper discusses a very interesting topic. However, some comments need to be addressed.
In the abstract is too long. Could you please reduce it?
Introduction:
Line 61 you stated, “Previous studies have found that long-term chronic exposure to high altitude (HA) 61 induces oxidative stress.” Could you please add references to these previous studies?
Similarly, in line 96 you stated, “So far, EEG studies on altitude hypoxia have mostly applied traditional statistical inference to compare the between-group differences with high- and low-altitude populations, which would be exposed to type I or type II errors. Could you please add more references to these studies?
Methods:
Why were all data collected from female participants?
What is the type and brand of EEG sensor used?
What are the inclusion and exclusion criteria for joining the experiment?
Usually, a notch filter should be used to eliminate powerline interference. Why didn’t the authors use it?
Deep learning techniques are state-of-the-art, I wonder why the authors didn’t use any of them?
In line 175, please verify the use of such an epoch size. Similarly, do the same for line 182.
All feature selection methods used in the manuscript are filter-based methods. they only rank features. Could you please mention the selection criterion to select among these rank features?
Experimental Results
Please define the metrics you are using for evaluating the performance of the classifiers and add their equations.
Statistical analysis should be performed for validating the machine learning classifiers' performance. please add either confident intervals or standard deviations.
Conclusion
Please add your future direction
Author Response
The paper aims to examine the effect of high altitude on inhibitory control processes that underlie sustained attention in the neural correlates of EEG data. Machine learning approaches involving EEG data of inhibitory control were used to classify high altitude chronic hypoxia and plain controls. The paper discusses a very interesting topic. However, some comments need to be addressed.
Answer: Special thanks to you for your good comments. Your comments and those of the reviewers were highly insightful and enabled us to greatly improve the quality of our manuscript. In the following pages are our point-by-point responses to each of the comments of the reviewers as well as your own comments.
Q1: In the abstract is too long. Could you please reduce it?
Response 1: Thanks for your suggestion! Considering your suggestion, we have reduced the Objective and Conclusions in Line 13-34:“(1) Objective: The aim of this study was to examine the effect of high altitude on inhibitory control processes that underlie sustained attention in the neural correlates of EEG data and explore whether the EEG data involving in inhibitory control contain valuable information to classify high-altitude chronic hypoxia and plain controls......4) Conclusions: Over other methods, the SVM leads to a good overall performance that increases with the time spent on task, illustrating that the ERPs and neural oscillations may provide neuroelectrophysiological markers for identifying chronic plateau hypoxia.”
Q2: Introduction:
Line 61 you stated, “Previous studies have found that long-term chronic exposure to high altitude (HA) 61 induces oxidative stress.” Could you please add references to these previous studies?
Similarly, in line 96 you stated, “So far, EEG studies on altitude hypoxia have mostly applied traditional statistical inference to compare the between-group differences with high- and low-altitude populations, which would be exposed to type I or type II errors. Could you please add more references to these studies?
Response 2: Thanks for your reminding! We have added references to these previous studies. Please see Line 59-60:“Previous studies have found that long-term chronic exposure to high altitude (HA) induces oxidative stress[23]”
Please see Line 95-97:“So far, EEG studies on altitude hypoxia have mostly applied traditional statistical inference to compare the between-group differences with high- and low-altitude populations, which would be exposed to type I or type II errors[36, 37].”
Methods:
Q3:Why were all data collected from female participants?
Response 3: Thanks for your question. All the data were collected from both male and female participants. We have added the number of male participants in Line 136-149. “A total of 35 healthy, right-handed college students (16 females, 19 males; 22.200 ± 1.699 years) in high-altitude areas (Lhasa: 3680 m) were recruited from the Tibet University campus for the present study. These participants lived in low-altitude areas (<500m) until early adulthood and had been living in Lhasa city for three years. We also collected data from 32 matched healthy college students (15 females, 17 males; 21.933 ± 1.438 years) in low-altitude areas (Guangzhou city) who had never been lived at high-altitude as the control group.”
Q4:What is the type and brand of EEG sensor used?
Response 4: Thanks for your question. All the data were collected from both male and female participants. We have added the number of male participants in Line 174 and 176. “Electroencephalograms (EEGs) were recorded from 64-channel Neuroscan SynAmps2 amplifier (10/20 system) using Ag/Agcl electrodes mounted on a Neuroscan 64-channel QuickCap (Compumedics USA, Charlotte, NC). ”
Q5:What are the inclusion and exclusion criteria for joining the experiment?
Response 5: Thanks for your question. Inclusion criteria for the plateau group (Lhasa: 3680 m) were as follows: these participants lived in low-altitude areas (<500m) until early adulthood and had been living in Lhasa city for three years. Exclusion criterion of high-altitude group was having a high altitude experience before attending college.
Inclusion criteria for the plain group (Guangzhou city) were as follows:Matched healthy college students in low-altitude areas (Guangzhou city) who had never been lived at high-altitude as the control group. Exclusion criterion of low-altitude group was having an experience of going to the plateau. All participants were right-handed, free of neurological and psychiatric diseases and had normal or corrected-to-normal vision.
Please see Line 136-152.
Q6:Usually, a notch filter should be used to eliminate powerline interference. Why didn’t the authors use it?
Response 6: Thanks for your question.We used a high-pass filter of 0.1 Hz (48 dB/oct) and a low-pass filter of 30 Hz (48 dB/oct). This filter can eliminate the 50 Hz interference. Please see the references:
- Clayson, P. E., Baldwin, S. A., Rocha, H. A., & Larson, M. J. (2021). The data-processing multiverse of event-related potentials (ERPs): A roadmap for the optimization and standardization of ERP processing and reduction pipelines. NeuroImage, 245, 118712.
- Kappenman, E. S., Farrens, J. L., Zhang, W., Stewart, A. X., & Luck, S. J. (2021). ERP CORE: An open resource for human event-related potential research. NeuroImage, 225, 117465.
Q7:Deep learning techniques are state-of-the-art, I wonder why the authors didn’t use any of them?
Response 7: Thanks for your question. Deep learning is a machine learning paradigm that
can be applied to the problems of prediction, forecasting, and classification. Deep learning is the subfield of machine learning concerned with extracting information from data in hierarchies
of concepts where more abstract concepts are built out of less abstract concepts (Urban & Gates, 2021). We mainly focus on whether the EEG data involving in inhibitory control contain valuable information to classify high-altitude chronic hypoxia and plain controls. So we used the three machine learning algorithms, namely a support vector machine (SVM), logistic regression (LR), and decision tree (DT). Nevertheless, Deep learning combined with Machine learning and deep learning may be a key step toward helping psychologists accurately and reliably classify high-altitude chronic hypoxia and plain controls.
[1] Urban, C. J., & Gates, K. M. (2021). Deep learning: A primer for psychologists. Psychological Methods.
Q8:In line 175, please verify the use of such an epoch size. Similarly, do the same for line 182.
Response 8: Thanks for your question. The epoch size of both the time domain analysis and Time–frequency analysis is a period of 1200ms (200ms baseline and 1000 ms post-stimulus onset). We have corrected this description in Line 195.“A time–frequency analysis was performed using the MATLAB toolbox FieldTrip [52] and epoched into a period of -200 to 1000 ms, corresponding to the onset of the word stimulus. ”
Q9:All feature selection methods used in the manuscript are filter-based methods. they only rank features. Could you please mention the selection criterion to select among these rank features?
Response 9: Thanks for your question. “The above two feature selection methods - ReliefF and GR were used for feature selection in the time and frequency domains, respectively. Each classifier extracted 36 features, including 12 time-domain and 24 frequency-domain features. All features generated by each classifier were consisted of two permutations and ranked from high to low. The first 18 features of the two permutations were selected as two datasets, and the intersection of the two datasets was used to obtain eight time-domain and three frequency-domain features”. Please see Line 316-322.
Experimental Results
Q10:Please define the metrics you are using for evaluating the performance of the classifiers and add their equations.
Response 10: Thanks for your suggestion. We have added the “The metrics used for evaluating the performance of the classifiers” in Line 358-364.
Table 1 The metrics used for evaluating performance of the classifiers
|
True value |
||
Positive |
Negative |
||
Predictive value |
Positive |
TP |
FN |
Negative |
FP |
TN |
Accuracy= (16)
F1score= (17)
Sensitivity= (18)
Specificity= (19)
AUC:The Area under the curve.
Q11:Statistical analysis should be performed for validating the machine learning classifiers' performance. please add either confident intervals or standard deviations.
Response 11: Thanks for your suggestion. We have add 95% confident intervals of Accuracy in Table 7.
Table 7 The values of precision, sensitivity, and specificity achieved from ERPs and EROs mode for Elman neural network by the SVM, LR, and DT algorithms |
||||||
Classifiers |
Periods |
Accuracy (95% confidence interval) |
F1score |
Sensitivity |
Specificity |
AUC |
SVM |
a1 |
86.57%(78.41%,94.73%) |
0.89 |
85.71% |
87.50% |
0.90 |
a2 |
88.06%(80.3%,95.82%) |
0.88 |
91.43% |
84.38% |
0.96 |
|
a3 |
92.54%(86.25%,98.83%) |
0.93 |
91.43% |
93.75% |
0.97 |
|
LR |
a1 |
95.52%(90.57%,100.00%) |
0.96 |
91.43% |
100.00% |
0.94 |
a2 |
89.55%(82.22%,96.88%) |
0.90 |
91.43% |
87.50% |
0.93 |
|
a3 |
86.57%(78.41%,94.73%) |
0.87 |
88.57% |
84.38% |
0.92 |
|
DT |
a1 |
80.60%(71.13%,90.07%) |
0.80 |
85.71% |
75.00% |
0.69 |
a2 |
70.15%(59.19%,81.11%) |
0.68 |
80.00% |
59.38% |
0.70 |
|
a3 |
70.15%(59.19%,81.11%) |
0.70 |
65.71% |
75.00% |
0.81 |
Conclusion
Q12:Please add your future direction
Response 12: Thanks for your suggestion. We have added our future direction as follows:“We should consider several issues in future works. First, the inter trial phase locking of neural oscillations characteristics of hypoxia tolerance, is a measure of phase consistency of the neural activities during experimental trails, and this would be a potential way to better understand the neural mechanisms of human adaptation to high altitudes. Second, because the amplitude of the EEG signal is really sensitive to the noise, a multiple linear regression model of the EEG power spectrum should be explored related to the role of sustained attention on the high-altitude chronic hypoxia. Third, deep learning should be combined with Machine learning. It may be a key step toward helping psychologists accurately and reliably classify high-altitude chronic hypoxia and plain controls.” Please see Line 596-605.

Reviewer 2 Report
Thank you for the opportunity to read your study about Event-Related EEG of sustained attention in high altitude versus low altitude individuals. I was impressed with how much detail you included in the Methods as well as Results. I do not have personal experience with EEG studies, so I found your detailed explanations very helpful. Your tables were informative without being over-crowded, and I appreciated the images you included.
There are a few small edits I detected in your paper:
--The sentence from line 80 to 84 is a very long sentence and is confusing to read. There is a lot of info in it that I think would be expressed better if the sentence was broken up into at least 2 sentences or shortened in some way with the details.
--In line 88, you have a period after et. that should not be there. Also need a space between et and al.
--In line 135, delete the word "been" to read..."who had never lived at high-altitude..."
--In line 286, is there a year or reference for the Robniksikonja and Kononenko reference?
I want to be sure I understand your results and conclusion. Unfortunately, the lines that differentiate HA versus LA in your graphs did not show up in my printed copy. So, individuals in HA had lower omission errors. I am putting myself in the role of someone without as much familiarity with sustained attention, etc, and thinking that I would read that and say to myself, "Well, then does that suggest those in HA have better functioning for sustained attention?" When I read your conclusion, I would think "And what does that mean for these individuals' everyday lives?" I don't think you need to go into too many details, but I think it would be helpful to make a statement or a few about this. This would allow your article to be understandable to more readers than if you leave it as is.
Author Response
Thank you for the opportunity to read your study about Event-Related EEG of sustained attention in high altitude versus low altitude individuals. I was impressed with how much detail you included in the Methods as well as Results. I do not have personal experience with EEG studies, so I found your detailed explanations very helpful. Your tables were informative without being over-crowded, and I appreciated the images you included.
Answer: Special thanks to you for your good comments. Your comments and those of the reviewers were highly insightful and enabled us to greatly improve the quality of our manuscript. In the following pages are our point-by-point responses to each of the comments of the reviewers as well as your own comments.
There are a few small edits I detected in your paper:
Q1:--The sentence from line 80 to 84 is a very long sentence and is confusing to read. There is a lot of info in it that I think would be expressed better if the sentence was broken up into at least 2 sentences or shortened in some way with the details.
Response 1: Thanks for your suggestion! Considering your suggestion, we have broken up the sentence into 2 sentences:"The GSAT is not only associated with top-down inhibitory control, but also explicitly reflects the distribution of attention resources over different periods of time. Higher omission errors during this task are associated with high-altitude hypoxia-related to the decline in the efficiency of cognitive control mechanisms [30] "
Q2:--In line 88, you have a period after et. that should not be there. Also need a space between et and al.
Response 2: Thanks for your reminding! We have made correction about this error and added a between et and al. in Line 87.
Q3:--In line 135, delete the word "been" to read..."who had never lived at high-altitude..."
Response 3: Thanks for your reminding! We have made correction about this sentence in Line 138. "These participants lived in low-altitude areas (<500m) until early adulthood and had never lived at high-altitude. Up to now, they had been living in Lhasa city for three years."
Q4:--In line 286, is there a year or reference for the Robniksikonja and Kononenko reference?
Response 4: Thanks for your reminding! There is a year “2003” for the Robniksikonja and Kononenko reference. We have added it in Line 286.
Robnik-Šikonja, M., & Kononenko, I. (2003). Theoretical and empirical analysis of ReliefF and RReliefF. Machine learning, 53(1), 23-69.
Q5:I want to be sure I understand your results and conclusion. Unfortunately, the lines that differentiate HA versus LA in your graphs did not show up in my printed copy. So, individuals in HA had lower omission errors. I am putting myself in the role of someone without as much familiarity with sustained attention, etc, and thinking that I would read that and say to myself, "Well, then does that suggest those in HA have better functioning for sustained attention?" When I read your conclusion, I would think "And what does that mean for these individuals' everyday lives?" I don't think you need to go into too many details, but I think it would be helpful to make a statement or a few about this. This would allow your article to be understandable to more readers than if you leave it as is.
Response 5: Thanks for your reminding! We have added the lines that differentiate HA versus LA in Figure 5. Additionally, we have simplified the conclusion and mainly reported the conclusions of EEG and machine learning as follows:
“Our work found that PFC cortical areas of HA individuals specific to the cognitive tasks were activated to compensate for reduced oxygen availability compared to low-altitude areas, which was reflected in the time domain (e.g., SP) and time-frequency domain (e.g., delta, slow alpha, fast alpha, and beta) datasets. Furthermore, data mining indicated that SVM is the most effective classification method to discriminate between chronic hypoxic exposure and a normoxic baseline at a later stage of sustained attention than the LR and DT algorithms. These findings not only provide novel insights into response inhibition in a situation of sustained attention in chronic exposure to high-altitude environments, but also reveal an auxiliary method for identifying chronic plateau hypoxia at the electroneurophysiological level.”
Please see Line 610-619.

Reviewer 3 Report
It has been a comprehensive study, and I think it has valuable content. The subject is fascinating. However, the following significant corrections seem necessary to improve the scientific level of the article.
1- What is the motivation for the work? The problem considered does not have a sound motivation. The authors should demonstrate a scientific interest in the objectives and results.
2- How did the authors optimize the number of features? Please use the hyper-optimization method based on the following article.
Automated detection of driver fatigue from electroencephalography through wavelet-based connectivity
3- Please show the distribution of extracted features by violin plot.
4- Is there any difference between the phases of neural data between the two groups? The reviewer suggests evaluating Inter-trial phase coherence in two groups as well.
5- The amplitude of the EEG signal is really sensitive to the noise. Please show the GLM of the EEG power spectrum based on the following article.
6- The logic of the introduction can be improved. For example, the reasons and significance of applying machine learning methods for behavioral-cognitive tasks. Current progress and critical issues could also be mentioned. The authors can use these articles to edit this section. A cognitive brain-computer interface monitoring sustained attentional variations during a continuous task, Frequency–amplitude coupling: a new approach for decoding of attended features in covert visual attention task, Decoding the status of working memory representations in preparation of visual selection, Behavioral decoding of working memory items inside and outside the focus of attention, Decoding covert visual attention based on phase transfer entropy
7- Please report the percentage of signal change in time courses of frequency band activity in recorded electrodes at each class (for more detail, please check the following article).
Beta‐band oscillations during passive listening to metronome sounds reflect improved timing representation after short‐term musical training in healthy older adults.
8- The conclusion should be more carefully rewritten, summarizing what has been learned and why it is interesting and valuable.
9- I would like to know more about the validation method. Do the authors leave one trial out of each subject, or they trained classifiers by using all trials from all subjects and then classifying one trial? Please report accuracy in each subject if the authors left one trial out.
Author Response
It has been a comprehensive study, and I think it has valuable content. The subject is fascinating. However, the following significant corrections seem necessary to improve the scientific level of the article.
Q1- What is the motivation for the work? The problem considered does not have a sound motivation. The authors should demonstrate a scientific interest in the objectives and results.
Response 1: Thanks for your suggestion! Considering your suggestion, we have added motivation for the work in the part of Abstract:“Objective: The aim of this study was to examine the effect of high altitude on inhibitory control processes that underlie sustained attention in the neural correlates of EEG data and explore whether the EEG data involving in inhibitory control contain valuable information to classify high-altitude chronic hypoxia and plain controls.......(3) Results: ...... Finally, the machine learning results showed that the SVM achieved the optimal classification f1 score in the latest stage of sustained attention, with an f1 score of 0.93, accuracy of 92.54%, sensitivity of 91.43%, specificity of 93.75%, and Area Under roc Curve (AUC) of 0.97. The results proved that SVM classifcation algorithms could be used to identify high-altitude chronic hypoxia. ”
2.How did the authors optimize the number of features? Please use the hyper-optimization method based on the following article.
Automated detection of driver fatigue from electroencephalography through wavelet-based connectivity
Response 2: In the above paper, the feature selection method adopts the recursive feature elimination method in the package method, and adopts the machine learning algorithm to evaluate the feature subset, which can detect the interaction relationship between two or more features. It is a method combining feature subset search and evaluation index. The effect of package method is the most conducive to improve the performance of the model among all the feature selection methods. It can use few features to achieve very good results.
However, the ReliefF and GainRatio methods were used in our study. These two methods are filtering methods and have the advantages of high computational efficiency without relying on any machine learning methods and requiring no cross-validation. The following two papers used ReliefF combined with machine learning methods for emotion recognition and epilepsy detection, respectively:
- Pippa, E., Zacharaki, E. I., Mporas, I., Tsirka, V., Richardson, M. P., Koutroumanidis, M., & Megalooikonomou, V. (2016). Improving classification of epileptic and non-epileptic EEG events by feature selection. Neurocomputing, 171, 576-585.
- Zhiguo Shi and Yu Cao (2016), ReliefF-Based EEG Sensor Selection Methods for Emotion Recognition, Sensors 2016, 16(10), 1558; doi:10.3390/s16101558
The following article uses GainRatio combined with machine learning methods to achieve automatic alcoholism detection:
[1] Shah, S., Sharma, M., Deb, D., Pachori, R.B. (2019). An Automated Alcoholism Detection Using Orthogonal Wavelet Filter Bank. In: Tanveer, M., Pachori, R. (eds) Machine Intelligence and Signal Analysis. Advances in Intelligent Systems and Computing, vol 748. Springer, Singapore. https://doi.org/10.1007/978-981-13-0923-6_41
Besides, in this paper, the authors designs a heuristic function to evaluate the attributes in node selection. This includes weighted information GainRatio, test costs, and user-specified non-positive parameters to adjust for the impact of test costs.
[1]Hong Zhao and Xiangju Li. 2017. A cost sensitive decision tree algorithm based on weighted class distribution with batch deleting attribute mechanism. Inf. Sci. 378, C (February 2017), 303–316. https://doi.org/10.1016/j.ins.2016.09.054
- Please show the distribution of extracted features by violin plot.
Response 3: Thanks for your suggestion! Considering your suggestion, we have added Figure 6. Violin diagrams.
Figure 6. Violin diagrams showed the distribution of the distribution of extracted features. GO-beta-max = Maximum power value of beta band during Go trials; NoGo-theta-max = Maximum power value of theta band during No-Go trials; NOGO-alpha1-max = Maximum power value of alpha1 band during No-Go trials; Go_mean = Average amplitude of SP components during Go trials; Go_Total area = Total area of SP components during Go trials; Go_|mean| = Absolute value of the mean amplitude of SP component during Go trials; Go_|latency/mean| = Absolute value of the ratio between latency and mean amplitude of SP components during Go trials; NoGo_mean = Average amplitude of SP components during No-Go trials; NoGo_Total area = Total area of SP components during No-Go trials; NoGo_|mean| = Absolute value of the mean amplitude of SP component during No-Go trials; NoGo_|latency/mean| = Absolute value of the ratio between latency and mean amplitude of SP components during No-Go trials.
- Is there any difference between the phases of neural data between the two groups? The reviewer suggests evaluating Inter-trial phase coherence in two groups as well.
Response 4: Thanks for your question! We compared the approaches to the classification of visual ERPs referring to Lina et al. (2021) and Li et al. (2018) .We agreed that comparison of phase coherence between groups could be considered in future work. Please see the future direction in Line 599-603: “First, the inter trial phase locking of neural oscillations characteristics of hypoxia tolerance, is a measure of phase consistency of the neural activities during experimental trails, and this would be a potential way to better understand the neural mechanisms of human adaptation to high altitudes. “
- Abou-Abbas, L., van Noordt, S., Desjardins, J. A., Cichonski, M., & Elsabbagh, M. (2021). Use of empirical mode decomposition in ERP analysis to classify familial risk and diagnostic outcomes for autism spectrum disorder. Brain Sciences, 11(4), 409.
- Li, X., Li, J., Hu, B., Zhu, J., Zhang, X., Wei, L., ... & Zhang, L. (2018). Attentional bias in MDD: ERP components analysis and classification using a dot-probe task. Computer methods and programs in biomedicine, 164, 169-179.
- The amplitude of the EEG signal is really sensitive to the noise. Please show the GLM of the EEG power spectrum based on the following article.
Response 5: Thanks for your suggestion! We compared the he EEG power spectrum based on the following article:
- Gutiérrez, D., & Ramírez-Moreno, M. A. (2016). Assessing a learning process with functional ANOVA estimators of EEG power spectral densities. Cognitive neurodynamics, 10(2), 175-183.
- Darracq, M., Sleigh, J., Banks, M. I., & Sanders, R. D. (2018). Characterising the effect of propofol on complexity and stability in the EEG power spectrum. British Journal of Anaesthesia, 121(6), 1368-1369.
We agreed that GLM of the EEG power spectrum could be considered in future work. Please see the future direction in Line 603-605: “Second, because the amplitude of the EEG signal is really sensitive to the noise, a multiple linear regression model of the EEG power spectrum should be explored related to the role of sustained attention on the high-altitude chronic hypoxia. ”
- The logic of the introduction can be improved. For example, the reasons and significance of applying machine learning methods for behavioral-cognitive tasks. Current progress and critical issues could also be mentioned. The authors can use these articles to edit this section. A cognitive brain-computer interface monitoring sustained attentional variations during a continuous task,2.Frequency–amplitude coupling: a new approach for decoding of attended features in covert visual attention task, 3.Decoding the status of working memory representations in preparation of visual selection, 4.Behavioral decoding of working memory items inside and outside the focus of attention, 5.Decoding covert visual attention based on phase transfer entropy
Response 6: Thanks for your suggestion! We have added the reasons, significance, current progress and critical issues of applying machine learning methods for behavioral-cognitive tasks. “Automatic classification of machine learning is an important step toward making EEG more practical and less dependent on trained professionals for applications. At the most basic level, EEG datasets consist of 2D (time and channel) matrices of real values representing brain-generated potentials recorded on the scalp in relation to specific task conditions. A great number of traditional machine learning algorithms such as Logistic Regression, SVM, Decision Tree and Random Forest have been applied on the EEG data. In the clinical, EEG signals combined with machine learning are used recently to identify sleep disorders, epilepsy, strokes,and other neurological disorders. Up to now, we only found a flight hypoxia detection system designed to classify normal from hypoxic using the transformed EEG data, which were performed on the Naïve Bayes, decision tree, random forest, and neural network algorithms, with sensitivity and specificity ranging from 0.83 ~ 1.00 and 0.91 ~ 1.00, respectively [38]. From a neuro-cognitive perspective, Knudsen’s attentional model proposed that inhibition of task-irrelevant processes played an vital role in sustained attention tasks. However, the classification performance of inhibitory control-related ERP components and neural oscillations in the GSAT as a tool for the auxiliary diagnosis of chronic altitude hypoxia stress has not been studied.”
7- Please report the percentage of signal change in time courses of frequency band activity in recorded electrodes at each class (for more detail, please check the following article).
Beta‐band oscillations during passive listening to metronome sounds reflect improved timing representation after short‐term musical training in healthy older adults.
Response 7: Considering the Reviewer’s suggestion, we have read this literature carefully. We found that “Time courses of induced beta activity in left and right auditory cortices for the 400-, 800- and 1200-ms beat conditions during pre- and post-training”. We have supposed that the researchers wanted to explore the sustained effect of auditory. So they analyzed the time courses of induced beta activity. However, we focused the the mean of frequency band activity within the time window of 460 to 700 ms same as the SP. Refer to the following literature, we just reported the mean effect of frequency band activity.
- Xiaowei Li , Jianxiu Li , Bin Hua, Jing Zhua, Xuemin Zhang,Liuqing Wei,Ning Zhong, et al. (2018) .Attentional bias in MDD: ERP components analysis and classification using a dot-probe task.Computer Methods and Programs in Biomedicine, 164 (2018): 169–179
- The conclusion should be more carefully rewritten, summarizing what has been learned and why it is interesting and valuable.
Response 8:Thanks for your suggestion! We have simplified the conclusion and mainly reported the conclusions of EEG and machine learning as follows:
“Our work found that PFC cortical areas of HA individuals specific to the cognitive tasks were activated to compensate for reduced oxygen availability compared to low-altitude areas, which was reflected in the time domain (e.g., SP) and time-frequency domain (e.g., delta, slow alpha, fast alpha, and beta) datasets. Furthermore, data mining indicated that SVM is the most effective classification method to discriminate between chronic hypoxic exposure and a normoxic baseline at a later stage of sustained attention than the LR and DT algorithms. These findings not only provide novel insights into response inhibition in a situation of sustained attention in chronic exposure to high-altitude environments, but also reveal an auxiliary method for identifying chronic plateau hypoxia at the electroneurophysiological level.”
- I would like to know more about the validation method. Do the authors leave one trial out of each subject, or they trained classifiers by using all trials from all subjects and then classifying one trial? Please report accuracy in each subject if the authors left one trial out.
Response 9: Thanks for your question! The validation method of leave one trial out of each subject: Assume that there are a total of K data in the data set, among which K-1 data is selected as the training set and the remaining 1 data is the test set to obtain the testing result. Next time, another data is selected as the testing set, and the remaining K-1 data is the training set to obtain the testing result. And so on, doing K validations until each data in the data serves as a test set. Finally, all the testing results are averaged to get the final accuracy. Since only one data was tested at a time, the results were only 0% or 100%. Due to the length limitation of the article, we only report the average classification results.
The classification of all subjects (1 = correct classification, 0 = wrong classification) |
|||||||||
Subject ID |
SVM |
LR |
DT |
||||||
a1 |
a2 |
a3 |
a1 |
a2 |
a3 |
a1 |
a2 |
a3 |
|
1 |
0 |
1 |
1 |
1 |
1 |
1 |
0 |
1 |
1 |
2 |
1 |
1 |
1 |
1 |
1 |
1 |
0 |
1 |
1 |
3 |
1 |
1 |
1 |
1 |
1 |
1 |
1 |
1 |
1 |
4 |
1 |
1 |
1 |
1 |
1 |
1 |
1 |
1 |
0 |
5 |
1 |
1 |
1 |
1 |
1 |
1 |
1 |
1 |
1 |
6 |
1 |
1 |
1 |
1 |
1 |
1 |
1 |
1 |
1 |
7 |
1 |
1 |
1 |
1 |
1 |
1 |
1 |
1 |
1 |
8 |
1 |
1 |
1 |
1 |
1 |
1 |
1 |
0 |
1 |
9 |
1 |
1 |
1 |
1 |
1 |
1 |
1 |
1 |
1 |
10 |
0 |
0 |
0 |
0 |
0 |
0 |
0 |
0 |
0 |
11 |
1 |
1 |
1 |
1 |
1 |
1 |
1 |
1 |
0 |
12 |
1 |
1 |
1 |
1 |
1 |
1 |
1 |
1 |
1 |
13 |
1 |
1 |
1 |
1 |
1 |
1 |
1 |
0 |
1 |
14 |
1 |
1 |
1 |
1 |
0 |
0 |
1 |
1 |
1 |
15 |
1 |
1 |
1 |
1 |
1 |
1 |
1 |
1 |
0 |
16 |
1 |
1 |
1 |
1 |
1 |
1 |
1 |
1 |
0 |
17 |
1 |
1 |
1 |
1 |
1 |
1 |
1 |
1 |
1 |
18 |
1 |
1 |
1 |
1 |
1 |
1 |
1 |
1 |
1 |
19 |
1 |
1 |
1 |
1 |
1 |
1 |
1 |
1 |
1 |
20 |
0 |
1 |
1 |
1 |
1 |
1 |
1 |
0 |
0 |
21 |
1 |
1 |
1 |
1 |
1 |
1 |
1 |
1 |
1 |
22 |
0 |
1 |
1 |
0 |
1 |
1 |
0 |
0 |
0 |
23 |
0 |
1 |
0 |
0 |
0 |
0 |
0 |
0 |
0 |
24 |
1 |
0 |
1 |
1 |
1 |
1 |
1 |
0 |
0 |
25 |
1 |
1 |
1 |
1 |
1 |
1 |
1 |
1 |
1 |
26 |
1 |
1 |
1 |
1 |
1 |
1 |
1 |
1 |
1 |
27 |
1 |
1 |
1 |
1 |
1 |
1 |
1 |
1 |
1 |
28 |
1 |
1 |
0 |
1 |
1 |
0 |
1 |
1 |
0 |
29 |
1 |
1 |
1 |
1 |
1 |
1 |
1 |
1 |
1 |
30 |
1 |
1 |
1 |
1 |
1 |
1 |
1 |
1 |
0 |
31 |
1 |
1 |
1 |
1 |
1 |
1 |
1 |
1 |
1 |
32 |
1 |
0 |
1 |
1 |
1 |
1 |
1 |
1 |
1 |
33 |
1 |
1 |
1 |
1 |
1 |
1 |
1 |
1 |
0 |
34 |
1 |
1 |
1 |
1 |
1 |
1 |
1 |
1 |
1 |
35 |
1 |
1 |
1 |
1 |
1 |
1 |
1 |
1 |
1 |
36 |
1 |
1 |
1 |
1 |
1 |
1 |
0 |
0 |
1 |
37 |
1 |
0 |
1 |
1 |
0 |
1 |
1 |
0 |
0 |
38 |
1 |
1 |
1 |
1 |
1 |
1 |
1 |
1 |
0 |
39 |
1 |
0 |
1 |
1 |
0 |
0 |
0 |
0 |
1 |
40 |
1 |
1 |
1 |
1 |
1 |
1 |
1 |
0 |
1 |
41 |
1 |
1 |
1 |
1 |
1 |
0 |
1 |
1 |
1 |
42 |
1 |
1 |
1 |
1 |
1 |
1 |
1 |
1 |
0 |
43 |
1 |
1 |
1 |
1 |
1 |
1 |
0 |
1 |
1 |
44 |
1 |
0 |
1 |
1 |
1 |
1 |
0 |
0 |
1 |
45 |
1 |
1 |
1 |
1 |
0 |
1 |
1 |
1 |
1 |
46 |
1 |
1 |
1 |
1 |
1 |
1 |
1 |
0 |
1 |
47 |
1 |
1 |
1 |
1 |
1 |
1 |
1 |
0 |
1 |
48 |
1 |
1 |
1 |
1 |
1 |
1 |
1 |
0 |
1 |
49 |
1 |
1 |
1 |
1 |
1 |
0 |
1 |
1 |
0 |
50 |
0 |
1 |
1 |
1 |
1 |
1 |
1 |
1 |
1 |
51 |
1 |
0 |
1 |
1 |
1 |
0 |
1 |
1 |
0 |
52 |
1 |
1 |
1 |
1 |
1 |
1 |
1 |
0 |
0 |
53 |
1 |
1 |
1 |
1 |
1 |
1 |
1 |
1 |
1 |
54 |
1 |
1 |
1 |
1 |
1 |
1 |
1 |
1 |
1 |
55 |
1 |
1 |
1 |
1 |
1 |
1 |
1 |
1 |
1 |
56 |
1 |
1 |
0 |
1 |
1 |
0 |
0 |
0 |
0 |
57 |
1 |
1 |
1 |
1 |
1 |
1 |
1 |
0 |
1 |
58 |
1 |
1 |
1 |
1 |
1 |
1 |
1 |
1 |
1 |
59 |
1 |
1 |
1 |
1 |
1 |
1 |
1 |
1 |
1 |
60 |
0 |
1 |
1 |
1 |
1 |
1 |
1 |
1 |
1 |
61 |
1 |
1 |
1 |
1 |
1 |
1 |
0 |
1 |
1 |
62 |
1 |
1 |
1 |
1 |
1 |
1 |
1 |
1 |
1 |
63 |
0 |
1 |
1 |
1 |
1 |
1 |
1 |
1 |
1 |
64 |
1 |
1 |
1 |
1 |
1 |
1 |
0 |
1 |
1 |
65 |
1 |
1 |
1 |
1 |
1 |
1 |
1 |
1 |
1 |
66 |
0 |
0 |
0 |
1 |
0 |
1 |
0 |
0 |
0 |
67 |
1 |
1 |
1 |
1 |
1 |
1 |
1 |
0 |
1 |

Round 2
Reviewer 1 Report
I would like to thank the reviewers for addressing most of my comments. The quality of the manuscript has improved. however, there are some comments that should be properly addressed.
1- please specify why did you use the epoch size of 1200 ms?
2- In the feature selection, we did you choose the first 18 highly ranked features?
Author Response
I would like to thank the reviewers for addressing most of my comments. The quality of the manuscript has improved. however, there are some comments that should be properly addressed.
1- please specify why did you use the epoch size of 1200 ms?
Response 1:Thanks for your question. According to previous studies about SP, we use the epoch size of 1200 ms. We have added the references into my manuscript.
- Chen X, Deng X. Differences in Emotional Conflict Processing between High and Low Mindfulness Adolescents: An ERP Study. International Journal of Environmental Research and Public Health. 2022; 19(5):2891. https://doi.org/10.3390/ijerph19052891
- Jia, L. X., Qin, X. J., Cui, J. F., Shi, H. S., Ye, J. Y., Yang, T. X., ... & Chan, R. C. (2021). Dissociation of proactive and reactive cognitive control in individuals with schizotypy: an event-related potential study. Journal of the International Neuropsychological Society, 27(10), 981-991.
- In the feature selection, we did you choose the first 18 highly ranked features?
Response 2:Thanks for your question. The features were ranked by the scores of the two methods (ReliefF and GainRatio) from the smallest to the largest, and then the images were generated from the two permutations. It can be seen that on the image corresponding to ReliefF, the 12th data started to approach 0 (Fig 1). On the image corresponding to information gain, the data starts to decrease from the 24th data (Fig 2). The two are averaged and the first 18 features are taken respectively.
Fig 1. ReliefF feature score trending chart
Fig 2. GainRatio feature score trending chart

Reviewer 3 Report
Accept.
Author Response
Thanks your comments again!